# Determination of Optimum Building Envelope Parameters of a Room concerning Window-to-Wall Ratio, Orientation, Insulation Thickness and Window Type

**Ayşe Fidan Altun**

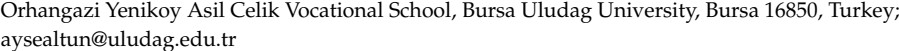

Orhangazi Yenikoy Asil Celik Vocational School, Bursa Uludag University, Bursa 16850, Turkey; aysealtun@uludag.edu.tr

**Abstract:** The building envelope includes all materials (glazing, external walls, doors, etc.) that separate the conditioned space from the outside environment. Building envelope characteristics significantly influence the energy consumption of buildings. In this study, research was carried out to find optimum building envelope design parameters, such as insulation thickness, orientation, glazing type, and the window-to-wall ratio of a room, using actual climatic data of two cities with different characteristics according to the Köppen climatic classification. The insulation thickness and the window type that minimizes the net present worth of the building façade over 20 years of a lifetime gave the optimum values. In addition, the effect of the various parameters, such as the infiltration rate through the envelope, room set-point temperature, and the fuel type, on the net present cost was also analyzed. It was found that appropriate selection of windows, orientation, and insulation thickness would lead to a significant reduction in the annual energy consumption. Despite having the lowest initial investment cost, the room with single glazed windows had the highest energy requirement and the net present cost. The building façade with double glazed windows, oriented towards the south-west, yielded the minimum net present cost in both locations. Results showed that the optimum external wall thickness is 9 cm in Hakkari (Dsa—Continental Climate) and 6 cm in Istanbul (Csa—Mild Climate).

**Keywords:** insulation thickness; energy efficiency; windows

## 1. Introduction

The energy required for buildings constitutes 40% of the total energy consumption globally [1]. A significant share of the energy consumed globally can be related to the heating demand, with a considerable amount associated with heat losses through the building envelope [2]. Recently, there has been an increasing interest in analyzing the thermal characteristics of the building envelope to decrease energy consumption. The building envelope is a crucial design subject to achieve indoor comfort levels with minimum energy requirements. As a result, it is imperative to consider all aspects, such as shape, orientation, climate, the envelope structure, and glazing type, from the beginning of a project. Proper design of the building envelope significantly decreases carbon emissions and energy consumption, and provides occupants with a healthy and comfortable indoor environment.

The appropriate design of the external walls can help achieve energy savings of around 50–60% [3]. Finding the optimum insulation thickness is part of this process. Due to the strong interest in reducing the energy use of the buildings, many researchers have focused on the optimization of insulation thickness. Many researchers have used a simplified equation by following the degree days method to minimize the net present value of the total costs, including of insulation and energy consumption. In a previous study, Kaynaklı presented a procedure to optimize thermal insulation thickness for external walls by considering costs of energy, insulation material, and installation. The study showed that the payback periods for the optimum insulation thickness vary between 3.85 and 16.25 years [4]. In another study,

Kaynaklı [5] investigated the influence of parameters such as building lifetime, inflation rate, cost of insulation material, thermal conductivity of the insulation material, and installation cost, on the total life cycle cost, energy savings rate and payback period. The results of the study showed that project lifetime, fuel cost, inflation rate, and thermal conductivity increase the optimum insulation thickness. In contrast, insulation material cost and Coefficient of Performance (COP) decrease the optimum insulation thickness. Kurekci [6] investigated the optimum insulation thickness for buildings with only heating, only cooling, and both heating and cooling energy requirements. Various insulation materials (extruded polystyrene, rock wool, expanded polystyrene, glass wool, and polyurethane) and different fuel types were used in the study. The calculations were undertaken for all city centers in Turkey. Ekici et al. conducted a study to investigate optimum insulation thickness for selected cities in the different climate regions by considering different wall types, different insulation materials, and different fuel types [7]. Based on the calculated optimum insulation thicknesses, energy savings and payback periods were presented for each selected location. Yuan et al. [8] investigated the optimum insulation thickness of external walls for 32 regions of China using the degree days method and life cycle cost analysis. Results of the study showed that 63% of the $CO_2$ emissions could be reduced when the optimum insulation thickness is applied. Canbolat et al. [9] investigated the optimum insulation thickness and payback period for two different climates. Using the Taguchi method, the importance order of the examined parameters was found, and the heating degree days was found to be the most efficient parameter based on the results. Alsayed and Tayeh [10] analyzed the optimum insulation thickness for Palestinian buildings considering weather data, insulation types, energy prices, and wall construction. Results of the study highlight the influence of degree days base temperature and insulation type on the optimum insulation thickness. Ozel et al. [11] investigated the optimum insulation thickness according to degree days, life cycle cost, and entransy loss methods. The calculations were undertaken for two different insulation materials. Acikkalp and Kandemir [12] presented a technique that combines economic and environmental effects to determine optimum insulation thickness. The proposed method is based on the degree day approach. A case study was carried out for the Bilecik province of Turkey. The optimum insulation thickness values were found to be between 0.13 and 0.47 m, depending on the environmental and economic priority. Barrau et al. [13] calculated the optimum insulation thickness considering different lifetimes of building and insulation materials. Results of the study show that changing the building lifetime from 20 to 50 years increases the optimum insulation thickness. Moreover, changing the optimization criteria from economic to environmental priority highly affects the results. Some researchers also combined life cycle assessment and exergy analysis to find the optimum insulation thickness [14–16]. To enhance the accuracy of predictions for the optimal thermal performance of the buildings, effective software programs were recommended to be used [17]. Simulation programs such as EnergyPlus [18–23] and TRNSYS [24,25] have been used by some researchers.

Windows also have a dominant role in energy consumption and the visual comfort of buildings. Finding the adequate window characteristics (such as orientation, area, and type of window) is part of the initial design decision, and is hard to change later [26]. Windows affects the energy requirement in different ways, such as heat conduction, solar radiation, and daily light transmission [27]. Windows can either reduce or increase the energy loads through solar heat gains or conduction heat losses. The energy transfer through windows depends on many parameters such as climatic conditions, shading levels, orientation, frame material, glazing type, area of the glazing, and many other factors. Detailed analysis must be undertaken to select the glazing types based on their impact according to the geographical location and respective climate conditions. There are only a few studies that have investigated the thermal performance of windows. Altun and Kılıç [26] presented a study to examine the influence of windows and shading device characteristics on the energy consumption of the buildings. Parameters such as the window-to-wall ratio of the façade, total solar energy transmittance value of the glazing, and shading levels regarding

orientations were studied. Gasparella [28] studied the impact of the glazing type, window size, and internal gains on the energy need of a residential building. Climatic data of Paris, Milan, Nice, and Rome were used in the research. Tsikaloudaki et al. [29,30] investigated the influence of the window configuration in terms of geometrical characteristics, thermophysical properties, orientation, and shading levels on its energy performance. Kon [31] conducted a study to investigate optimum insulation thickness and glass number for different climatic conditions, fuel types, and insulation materials. Ozel [32] conducted a study to examine the effect of the window-to-wall ratio on the optimum insulation thickness considering wall orientations. Results of the study showed that the window area and orientation have a significant influence on the optimum insulation thickness. Karabay and Arıcı [33] optimized multi-pane windows for different locations using the degree day method. The optimum number of the glazing for each selected location was found by considering the life cycle cost. Derradji et al. [34] conducted a study to investigate the optimum insulation thickness of a prototype building both numerically and experimentally. Various parameters such as wall structure, window area, and fuel type were studied.

In the literature, previous studies primarily focus on optimizing the insulation thickness based on heating degree days or cooling degree days. Although it is straightforward and fast to estimate, the degree days concept has been criticized for not considering the solar radiation and thermal mass effect [35]. Various researchers have questioned the accuracy of the method. It is essential to consider the building envelope as a whole when conducting optimization studies. Only a handful of studies have considered the building façade as a whole (windows and external walls together). Previous studies have mainly been conducted separately for exterior walls or windows. To the best of the author's knowledge, no study has simultaneously focused on insulation thickness, window-to-wall ratio, orientation, different climatic conditions, and window type of the building envelope.

In this study, a thermo-economic analysis of a designed zone was conducted for two selected cities in Turkey, which belong to different climate zones according to the Köppen climatic classification. The dynamic energy behavior of the zone was simulated for a year. Net present cost analyses were used for economic calculations. A comprehensive parametric evaluation was undertaken by varying the insulation thickness of the external wall, glazing type, glazing area, infiltration rate, room set-point temperature, fuel type, and wall orientation. The dynamic simulation results yielded essential insights for both energy savings and minimizing the net present cost. The methodology used in the study can be applied by other researchers, engineers, and architects around the world to design both energy-efficient and cost-effective buildings.

## 2. Materials and Methods

### 2.1. Description of the Reference Zone

Parametric evaluation was carried out for a reference room. The plan of the room is shown in Figure 1.

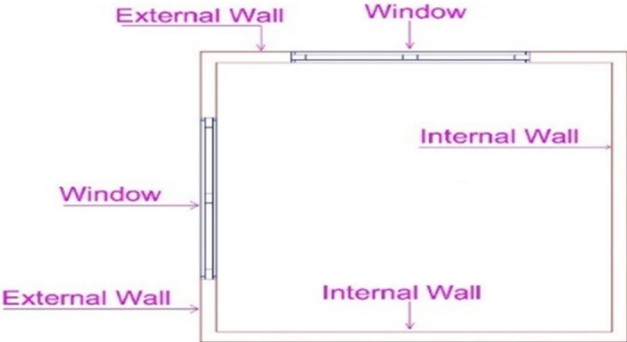

**Figure 1.** The layout of the reference room.

The designed zone has two external walls facing two different orientations. The room has a 100 m² floor area. The windows are considered to be located at the center of each exterior wall of the room. The dimensions of the room are 10 m (l), 10 m (w), 3 m (h). External walls are insulated, and insulation thickness is an investigation parameter. The properties of the building components are presented in Table 1. Three different glazing types (single, double, and triple glazed) were used to determine the optimum one. The heating temperature set-points were assumed to equal to 24 °C for the base case; however, this was also investigated parametrically. Occupant density was considered equivalent to 0.1 occupants/m², and specific lighting gains were determined as 10 W/m² during occupied hours if the total horizontal radiation level was lower than 120 W/m². For the base case, the infiltration rate was considered to be 0.2 ACH (Air Change per Hour); this was also investigated parametrically. In Table 1, the thermal properties of the external wall are given. The main design parameters of the zone are shown in Table 2. Three types of windows were investigated: (1) single glazed window, (2) double glazed window, (3) triple glazed window with argon filling. Four different façade orientations were analyzed as (1) south-east, (2) south-west, (3) north-east, (4) north-west.

**Table 1.** Thermal properties of the external wall.

| Material | Thickness (m) | Conductivity $(kJ \cdot h^{-1} m^{-1} K^{-1})$ | Density (kg·m$^{-3}$) |
|---|---|---|---|
| Plaster | 0.020 | 5 | 2000 |
| Brick | 0.210 | 3.2 | 1800 |
| Plaster | 0.030 | 5 | 2000 |
| Insulation | 0.03–0.15 | 0.144 | 40 |

**Table 2.** Main design parameters of the room.

| Parameter | Value |
|---|---|
| Area | 100 m² |
| Height | 3 m |
| Window Type | Single glazed (5.69 W/m²·K) |
| | Double glazed (1.1 W/m²·K) |
| | Triple glazed (0.61 W/m²·K) |
| Infiltration rate | 0.2–0.4–0.6–0.8–1 ACH |
| Orientation | North-east, north-west |
| | South-east, south-west |
| Heating set-point temperature | 18–20–22–24–26 °C |

### 2.2. Model Design and Calculation Methods

The degree day method is a widespread method for assessing and classifying climate regions with common climatic characteristics. In the literature, most of the studies have used this method to calculate optimum insulation thickness [9,33,36–38]. However, energy consumption in buildings depends on many parameters, such as building occupancy, solar radiation, equipment usage, and infiltration rate. Therefore, to find optimum insulation thickness, all these aspects should be considered to obtain realistic results.

In this study, dynamic and transient analysis was conducted using a simulation tool. TRNSYS 18 Simulation software was used for the thermal model. The TRNSYS program is a widely-known energy analysis program primarily used by researchers [39]. The TRNBuild interface of the program was used to create the building. TRNBuild allows the user to define building construction layers and create various infiltration and ventilation types

and occupancy schedules. For the meteorological data, the Meteonorm database of the program library was used. Typical meteorological year (TMY) weather data for the selected locations were used.

Space heating and cooling loads have four major components; solar heat gains through apertures, heat conduction, ventilation/infiltration, and internal loads [40]. The heat transfer between the building envelope, and outside and inside environments, can be described by conduction, convection, and radiation mechanisms. Convection heat flux $Q_i$ to a zone due to the difference between the indoor and outdoor temperatures can be expressed as:

$$Q_i = Q_{surf} + Q_{inf} + Q_{ven} + Q_g + Q_{cpl} + Q_{sol} + Q_{ISH} \tag{1}$$

The infiltration gains/losses ($Q_{inf}$) are expressed as below:

$$Q_{inf} = V \cdot \rho \cdot c_p \cdot (T_{out} - T_{air}) \tag{2}$$

In Equation (1), $Q_{ven}$ is the ventilation gains/losses, $Q_g$ and $Q_{cpl}$ are the internal convective gains (by people, equipment, illumination etc.) and gains due to connective air from the boundary condition, respectively.

$$Q_{ven} = V \cdot \rho \cdot c_p \cdot (T_{ven} - T_{air}) \tag{3}$$

where $\rho$ is the air density (kg/m$^3$), $c_p$ is the air specific heat (kJ/kg·K), V is the airflow rate (m$^3$/s). $Q_{sol}$ is the fraction of solar radiation entering a building zone through external windows that transfer as a convective gain to the inside air. $Q_{ISH}$ is the absorbed solar radiation on all internal shading devices that is directly transferred to the inside air.

### 2.3. System Performance Evaluation Parameters

It is apparent that as the insulation thickness increases, the cost of insulation also increases, and that the energy cost decreases. Similarly, single pane windows have a lower investment cost than double glazed and triple glazed windows. However, the application of windows with lower energy transmittance values decreases the annual energy cost. Therefore, it is essential to consider the net present worth of the building envelope that considers the initial investment cost and annual energy cost over the lifetime. In Equation (4), the investment cost of the external wall insulation is given:

$$C_{ins} = C_i \times x \times A_w \tag{4}$$

In Equation (4), $C_i$ is the cost of the insulation material per volume ($/m$^3$), $x$ is the insulation thickness (m), and $A_w$ is the wall area without the glazing (m$^2$). Karabay and Arıcı [33] obtained manufacturer prices and correlated the cost of the multiple panes. The investment cost of the multi-pane window ($C_I$) per unit is given below [33]:

$$C_I = (19.25 \times n) + 49 \tag{5}$$

In Equation (5), $n$ is the number representing the glazing. The total investment cost of a multi-pane window $C_w$ can be calculated by multiplying $C_I$ with the glazing area $A_g$ as below:

$$C_w = C_I \times A_g \tag{6}$$

Assuming $C_F$ is the unit price of the fuel, $LHV$ is the lower heating value of the fuel, and $\eta_b$ is the efficiency of the heating equipment, the annual heating cost of the designed zone ($C_h$) can be calculated as below:

$$C_h = \frac{Q_h}{LHV \times \eta} \times C_F \tag{7}$$

$Q_h$ is the annual heating load, which is integrated for a year. For $N$ years of the project lifetime, the total heating cost can be determined by multiplying Equation (7) with the present worth factor (*PWF*), which is given as:

$$PWF = \frac{(1+r)^N - 1}{r(1+r)^N} \tag{8}$$

In Equation (8), $i$ is the interest rate, and $g$ is the inflation rate; for $i > g$, $r$ can be written as:

$$r = \frac{i-g}{1+g} \tag{9}$$

For $i < g$:

$$r = \frac{g-i}{1+i} \tag{10}$$

For $i = g$:

$$PWF = \frac{N}{1+i} \tag{11}$$

To find the optimum thickness of thermal insulation, the insulation of the external walls of the façade and the window were considered to be an investment [37]. The total net present cost of the building envelope can be obtained as below:

$$C_t = C_h \times PWF + C_{ins} + C_w \tag{12}$$

To investigate energy savings due to insulation, the energy savings rate parameter is used. Energy savings rate can be defined as below:

$$E = \frac{Q_{h_{un}} - Q_{h_{ins}}}{Q_{h_{un}}} \tag{13}$$

In Equation (13), $Q_{h_{un}}$ is the annual heating energy requirement of the zone with uninsulated external walls. $Q_{h_{ins}}$ is the annual heating energy requirement with external wall insulation. Energy savings rate changes between 0 and 1, and greater values show better energy efficiency.

The economic predictions were made using published data from the Central Bank of Turkey [41] and the National Institute of Statistics [42]. All of the parameters selected in this study are given in Table 3. The flow chart of the simulation process that was used for all simulation cases is depicted in Figure 2.

**Table 3.** Parameters for the financial analysis.

| Parameter | Value [1] |
|---|---|
| Natural Gas | Unit price: 0.18 $/m$^3$ [43] |
| | *LHV*: 9.59 kWh/m$^3$ |
| | η : 98% |
| Coal | Unit price: 0.13 $/m$^3$ [43] |
| | *LHV*: 5.76 kWh/kg |
| | η : 65% [7] |
| Liquid Petrol Gas (LPG) | Unit price: 1.60 $/kg [43] |
| | *LHV*: 12.9 kWh/kg |
| | η : 90% [7] |
| Electricity | Unit price: 0.15$/kWh [43] |
| | *LHV*: 1 kWh/kWh |
| | η : 99% |

**Table 3.** *Cont.*

| Parameter | Value [1] |
|---|---|
| | Unit price: 1.03 $/kg [43] |
| Fuel-Oil | *LHV*: 11.28 kWh/kg |
| | η : 80% [44] |
| Interest rate (i) | 15% |
| Inflation rate (g) | 14.6% |
| Project lifetime | 20 years [45,46] |
| The unit price of the selected insulation material | 100 $/m³ [12] |

[1] TL/USD currency conversion set at 01.03.2022 1 $ = 13.93 TL.

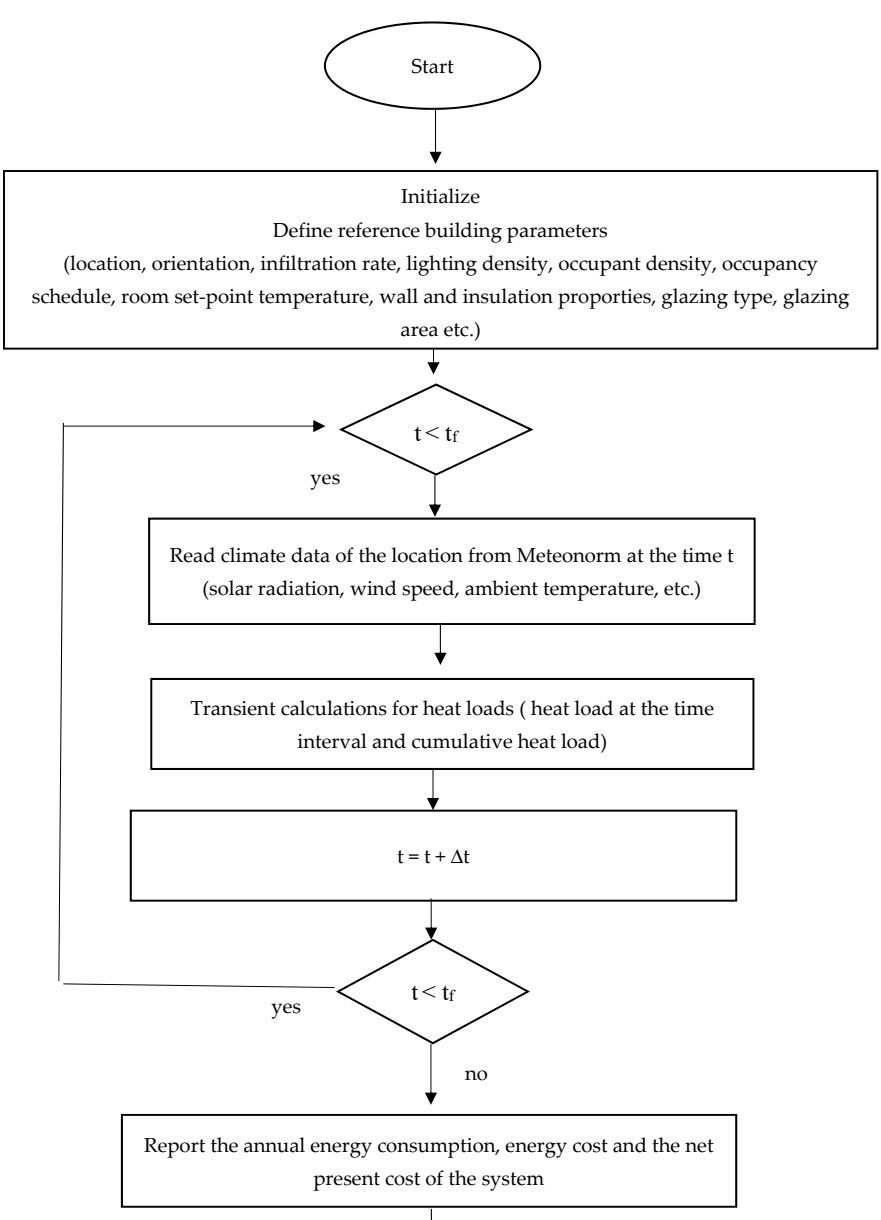

**Figure 2.** The flow chart of the simulation process.

### 3. Analysis

Turkey is positioned at the intersection of Europe, Asia, and the Middle East. It experiences different climatic conditions [47] and is divided into four climate zones. Two representative cities from different zones were selected to show the influence of the climatic conditions on optimum building envelope parameters. In Table 4, geographic information is presented for the selected cities. The heating degree days of Istanbul and Hakkari are 1860 and 3470, respectively [4], whereas cooling degree days are only 6 and 18. Both locations are heating-dominated; therefore, thermo-economic optimization of the building envelope was conducted based on the heating load. The Köppen climate classification system is widely used to classify the climate of a region. This classification has five major climate groups, which are A (tropical), B (dry), C (mild), D (continental), and E (polar). Each major group is divided into sub-groups. As shown in Table 4, based on this classification, Istanbul is an example of the Mild Climate (Csa), and Hakkari is an example of the Continental Climate (Dsa). In Figures 3 and 4, the monthly minimum, maximum, and mean ambient temperatures of the selected locations are presented.

**Table 4.** Climatic zones, topographic features, and degree days of the selected locations.

| Selected City | Istanbul | Hakkari |
|---|---|---|
| TS 825 Climate Zone | 2 | 4 |
| Latitude | 41°00′ N | 37°44′ N |
| Longitude | 28°97′ E | 43°74′ E |
| Altitude (Elevation) | 40 m | 1728 m |
| HDD (Heating Degree Days) | 1865 | 3470 |
| CDD (Cooling Degree Days) | 6 | 18 |
| Köppen Classification Major group | C (Mild) | D (Continental) |
| Köppen Classification Sub group | Csa | Dsa |

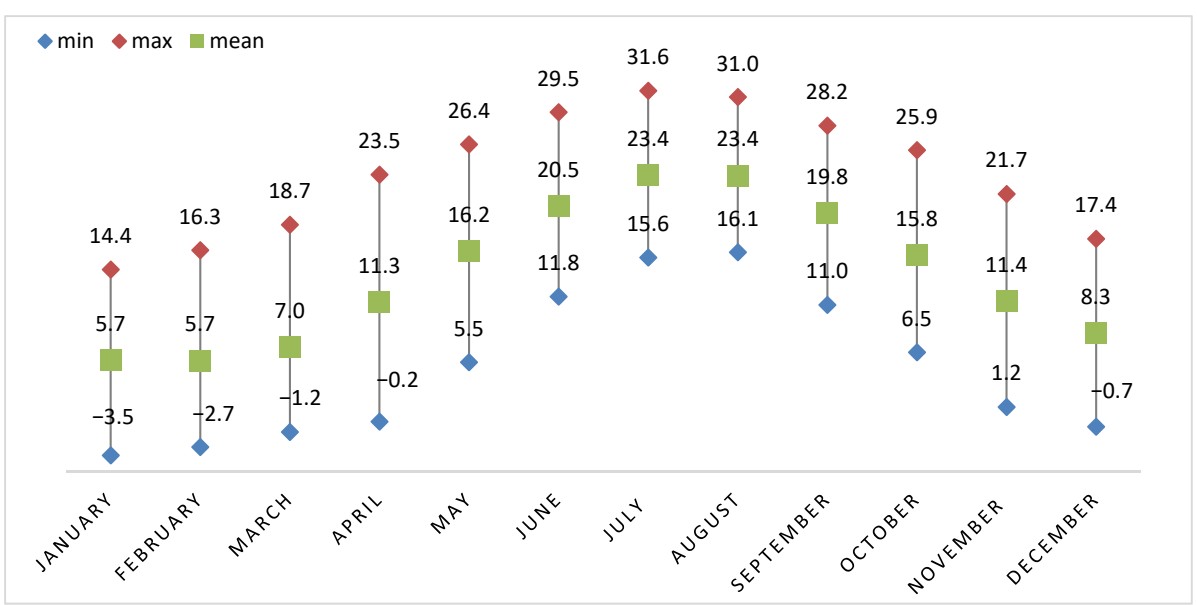

**Figure 3.** Mean, minimum, and maximum monthly temperatures for Istanbul.

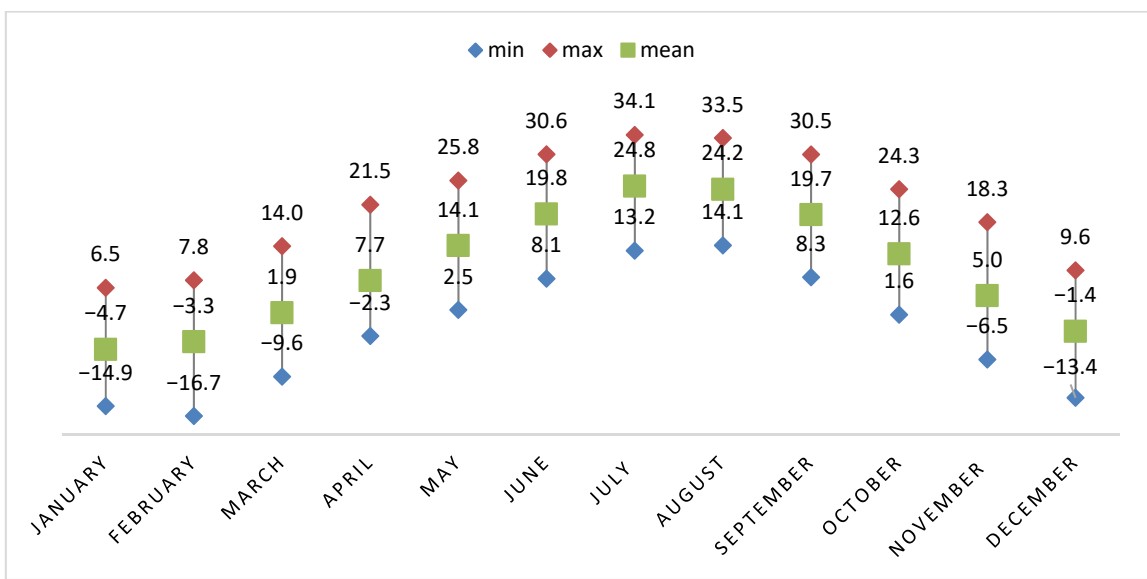

**Figure 4.** Mean, minimum, and maximum monthly temperatures for Hakkari.

In this study, the influence of insulation thickness, glazing type, infiltration rate, fuel type, room set-point temperature, window-to-wall ratio, and wall orientation on annual heating energy requirement, investment cost, and the net present worth was investigated. Results revealed information about design of a building façade with the minimum energy requirement. Table 5 presents all of the studied cases.

**Table 5.** Investigated cases.

| Case | Insulation Thickness (cm) | Glazing Type | Window to Wall Ratio | Orientation | Infiltration Rate | Heating Set-Point | Fuel Type |
|------|---------------------------|--------------|----------------------|-------------|-------------------|-------------------|-----------|
| 1 | 3 | Single/Double/Triple | 40% | NW | 0.2 ACH | 24 °C | Natural Gas |
| 2 | 6 | Single/Double/Triple | 40% | NW | 0.2 ACH | 24 °C | Natural Gas |
| 3 | 9 | Single/Double/Triple | 40% | NW | 0.2 ACH | 24 °C | Natural Gas |
| 4 | 12 | Single/Double/Triple | 40% | NW | 0.2 ACH | 24 °C | Natural Gas |
| 5 | 15 | Single/Double/Triple | 40% | NW | 0.2 ACH | 24 °C | Natural Gas |
| 6 | 3 | Single/Double/Triple | 40% | NE | 0.2 ACH | 24 °C | Natural Gas |
| 7 | 3 | Single/Double/Triple | 40% | NW | 0.2 ACH | 24 °C | Natural Gas |
| 8 | 3 | Single/Double/Triple | 40% | SE | 0.2 ACH | 24 °C | Natural Gas |
| 9 | 3 | Single/Double/Triple | 40% | SW | 0.2 ACH | 24 °C | Natural Gas |
| 10 | 3 | Single/Double/Triple | 20% | NW | 0.2 ACH | 24 °C | Natural Gas |
| 11 | 3 | Single/Double/Triple | 40% | NW | 0.2 ACH | 24 °C | Natural Gas |
| 12 | 3 | Single/Double/Triple | 60% | NW | 0.2 ACH | 24 °C | Natural Gas |
| 13 | 3 | Single/Double/Triple | 80% | NW | 0.2 ACH | 24 °C | Natural Gas |
| 14 | 3 | Single/Double/Triple | 100% | NW | 0.2 ACH | 24 °C | Natural Gas |
| 15 | 3 | Single/Double/Triple | 40% | NW | 0.2 ACH | 24 °C | Natural Gas |
| 16 | 3 | Single/Double/Triple | 40% | NW | 0.4 ACH | 24 °C | Natural Gas |
| 17 | 3 | Single/Double/Triple | 40% | NW | 0.6 ACH | 24 °C | Natural Gas |
| 18 | 3 | Single/Double/Triple | 40% | NW | 0.8 ACH | 24 °C | Natural Gas |

**Table 5.** *Cont.*

| Case | Insulation Thickness (cm) | Glazing Type | Window to Wall Ratio | Orientation | Infiltration Rate | Heating Set-Point | Fuel Type |
|------|---------------------------|--------------|----------------------|-------------|-------------------|-------------------|-----------|
| 19 | 3 | Single/Double/Triple | 40% | NW | 1.0 ACH | 24 °C | Natural Gas |
| 20 | 3 | Single/Double/Triple | 40% | NW | 0.2 ACH | 18 °C | Natural Gas |
| 21 | 3 | Single/Double/Triple | 40% | NW | 0.2 ACH | 20 °C | Natural Gas |
| 22 | 3 | Single/Double/Triple | 40% | NW | 0.2 ACH | 22 °C | Natural Gas |
| 23 | 3 | Single/Double/Triple | 40% | NW | 0.2 ACH | 24 °C | Natural Gas |
| 24 | 3 | Single/Double/Triple | 40% | NW | 0.2 ACH | 26 °C | Natural Gas |
| 25 | 3 | Single/Double/Triple | 40% | NW | 0.2 ACH | 24 °C | Natural Gas |
| 26 | 3 | Single/Double/Triple | 40% | NW | 0.2 ACH | 24 °C | Fuel-Oil |
| 27 | 3 | Single/Double/Triple | 40% | NW | 0.2 ACH | 24 °C | Coal |
| 28 | 3 | Single/Double/Triple | 40% | NW | 0.2 ACH | 24 °C | LPG |
| 29 | 3 | Single/Double/Triple | 40% | NW | 0.2 ACH | 24 °C | Electricity |

## 4. Results and Discussion

### 4.1. Orientation

In this part of the study, the reference room was rotated in such a way that exterior walls faced one of the four main orientations. The two outer walls were changed to meet north-east (NE), south-east (SE), north-west (NW), and south-west (SW) directions. In Figure 5, the effect of façade orientation on the net present cost is given. Results revealed that the net present cost of the system is minimized for the façade with double glazed windows, oriented towards the south-west. Despite having the lowest investment cost, the façade with single glazed windows has the highest net present cost values due to the high annual heating energy requirement. The net present cost can be cut by 35% if the façade is oriented towards the south-west instead of the north-east, and a double glazed window is selected instead of the single glazed window.

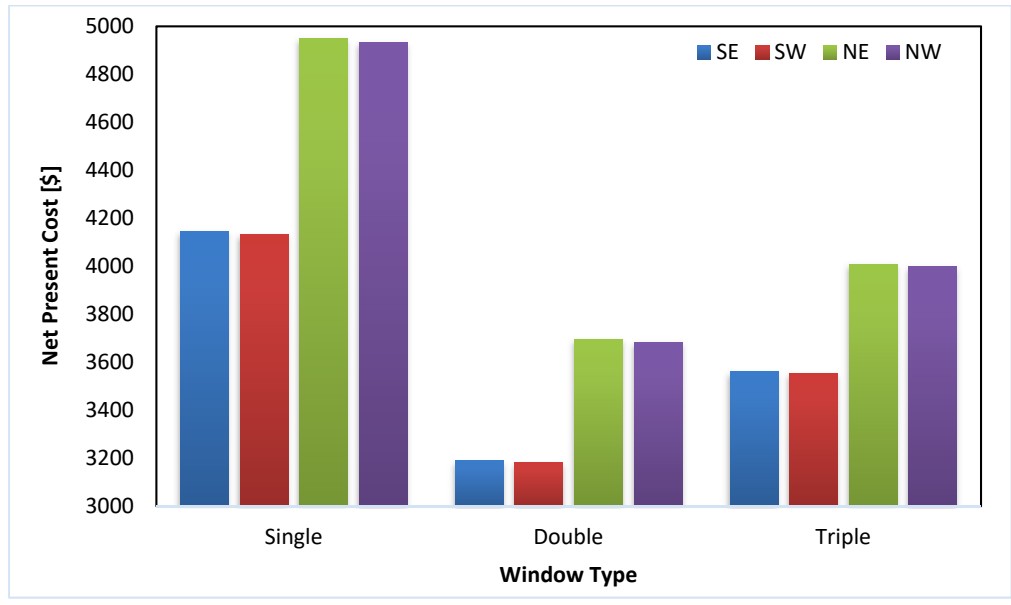

**Figure 5.** The influence of orientation on the net present cost of the building façade in Istanbul.

### 4.2. Window-to-Wall Ratio (WWR)

Windows are key design elements in architectural applications because they improve the appearance of buildings and enable daylight penetration [48]. Therefore, the effect

of the glazing area on the thermal performance of the buildings is an important research goal. A parametric evaluation was conducted to investigate various window-to-wall ratios (20%, 40%, 60%, 80%, 100%) according to be single glazed, double glazed or triple glazed windows. In Figures 6 and 7, the effect of the WWR on the annual heating cost is presented for Istanbul and Hakkari, respectively. Results revealed that the annual heating energy consumption of the room façade with a single glazed window increases with increasing window area. Due to lower energy transmittance values, the annual heating energy cost decreases with growing window area for double glazed and triple glazed façades. Moreover, the influence of the WWR on the heating energy cost is more dramatic in Hakkari than in Istanbul due to colder climatic conditions throughout the winter.

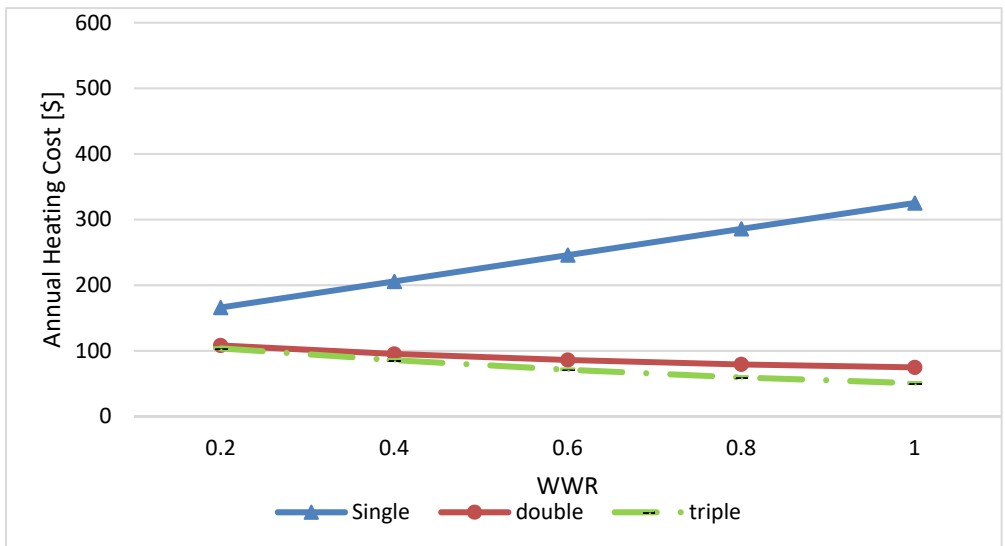

**Figure 6.** The influence of the window-to-wall ratio on annual heating cost in Istanbul (3 cm insulation, NW oriented).

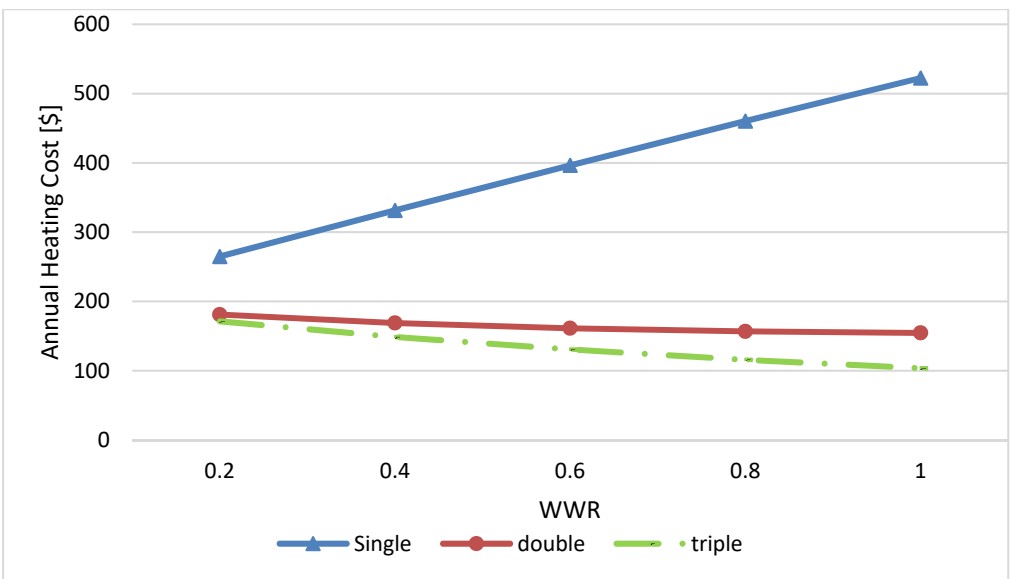

**Figure 7.** The influence of the window-to-wall ratio on the annual heating cost in Hakkari (3 cm insulation, NW oriented).

In Figures 8 and 9, the effect of the window-to-wall ratio and the glazing type on the net present cost of the system is presented. Results show that the net present cost increases with increasing window area. A fully glazed façade with a single pane window has the highest net present cost, of USD 9134 and 12,189 in Istanbul and Hakkari, respectively. A

double glazed façade with a 20% window-to-wall ratio has the lowest net present cost value in Istanbul and Hakkari of USD 2870 and 4000, respectively. Although increasing the window surface area increases the net present cost, window sizes cannot be reduced without considering visual comfort.

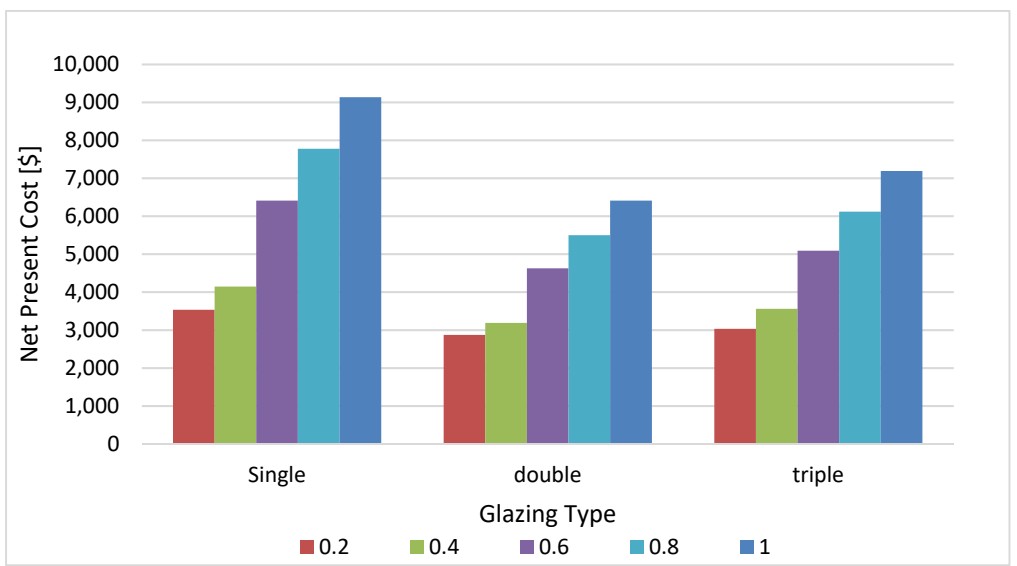

**Figure 8.** The influence of the window-to-wall ratio and the glazing type on the net present cost of the system in Istanbul (3 cm insulation, NW oriented).

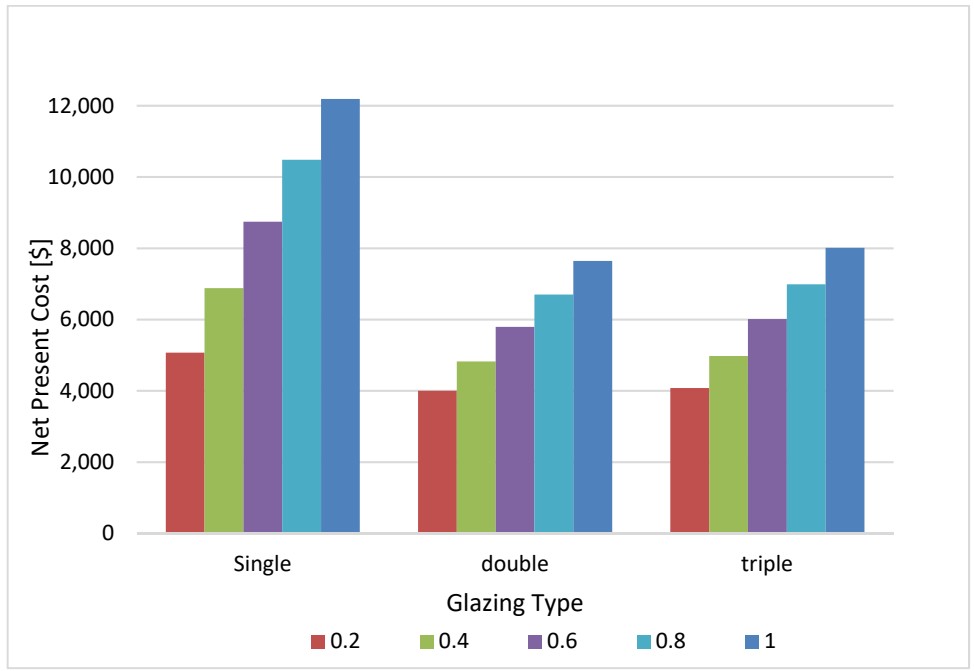

**Figure 9.** The influence of the window-to-wall ratio and the glazing type on the net present cost of the system in Hakkari (3 cm insulation, NW oriented).

In Figure 10, the influence of the window-to-wall ratio, orientation, and the glazing type on the net present cost of the façade are shown. Increasing the window area increases the net present value of the façade for both south-west and north-west directions. However, due to passive heating and higher solar radiation levels in south-oriented façades, the net present cost is always lower than that of the north-oriented façades.

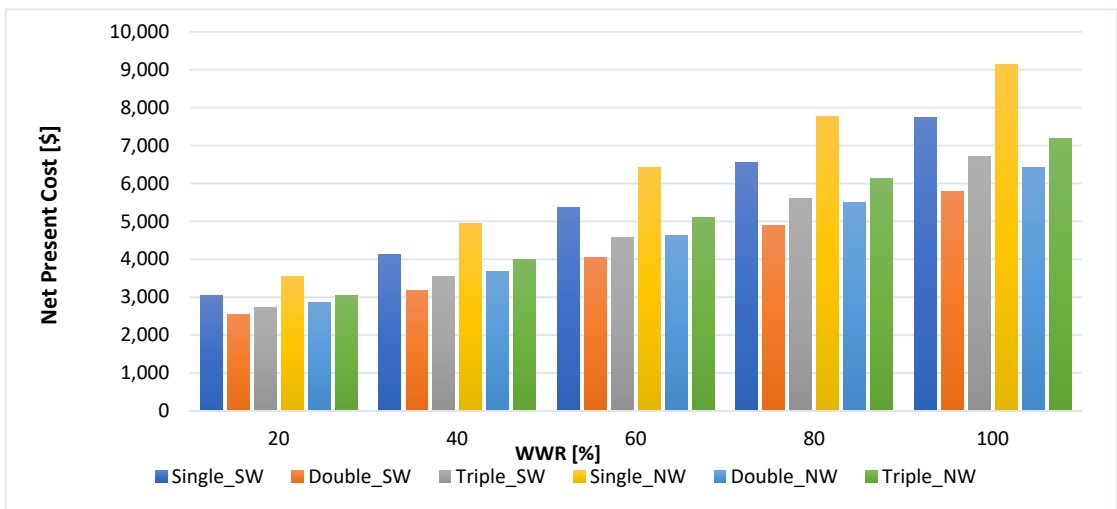

**Figure 10.** The influence of the window-to-wall ratio, orientation, and the glazing type on the net present cost of the system in Istanbul (3 cm insulation).

### 4.3. Insulation Thickness

In this part of the study, energy consumption and the net present cost were calculated by applying various insulation thicknesses of the external walls and window types. Optimum insulation thickness was obtained for which the net present cost value was minimized. In Figure 11, the influence of the insulation thickness on the net present cost of the system in Hakkari and Istanbul is shown. Results show that from 3 to 9 cm insulation thickness, the net present cost of the zone decreases dramatically for the façades with both double glazed and triple glazed windows. It can be seen that choosing a thickness value greater than 9 cm will increase the net present cost in Hakkari; therefore, it is unnecessary. The minimum net present cost was found to be USD 4584 for the façade with double glazed windows and 9 cm external wall insulation thickness in Hakkari. The optimum insulation thickness was found to be 6 cm for single, double, and triple glazed windows in Istanbul. The minimum net present cost was found to be USD 3583 for the façade with double glazed windows and 6 cm external wall insulation thickness.

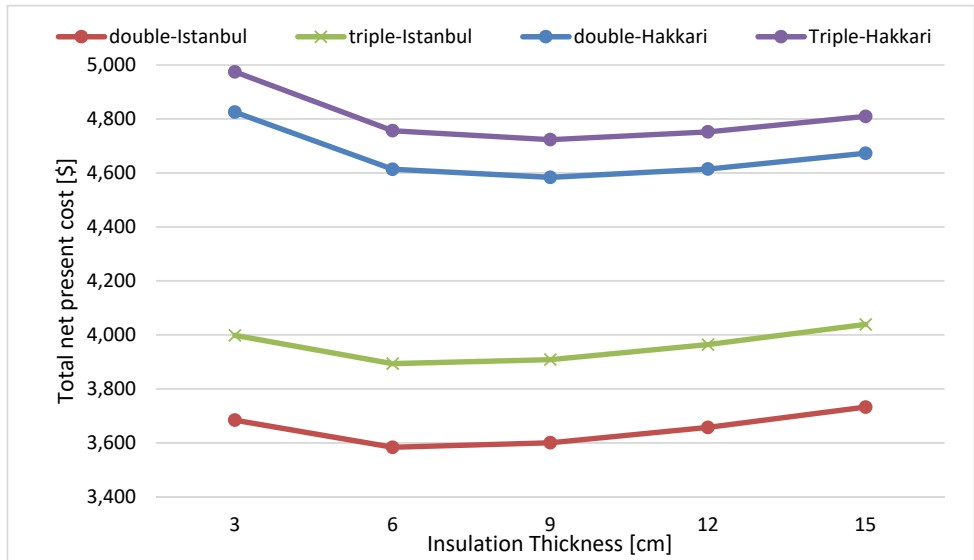

**Figure 11.** The influence of insulation thickness on the net present cost (40% WWR, NW oriented façade) for the selected locations.

Figure 12 demonstrates the investment cost of the façade for different insulation thicknesses and window types. It is seen that the investment cost increases with the increase in insulation thickness. In addition, a single glazed window façade has the minimum, and the triple glazed façade has the maximum investment cost.

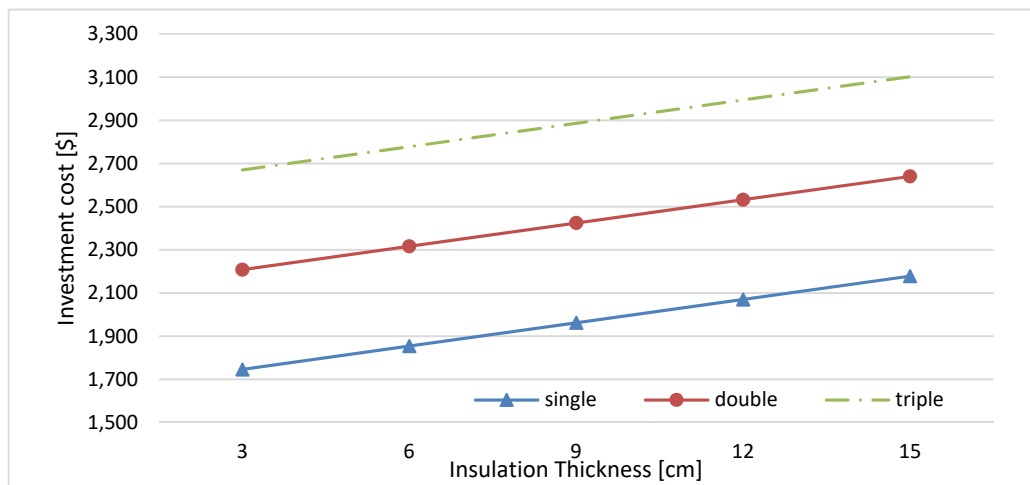

**Figure 12.** Effect of insulation thickness and window type on the insulation cost of the façade.

Figure 13 shows the annual natural gas consumption of the zone versus different insulation thicknesses and window types. Results show that for the façade with a single glazed window, changing the insulation thickness from 3 to 15 cm decreases the amount of fuel consumption by around 10–12%. For the double glazed and triple glazed façades, increasing the insulation thickness from 3 to 15 cm decreases the fuel consumption by 22–26% and 25–30%, respectively.

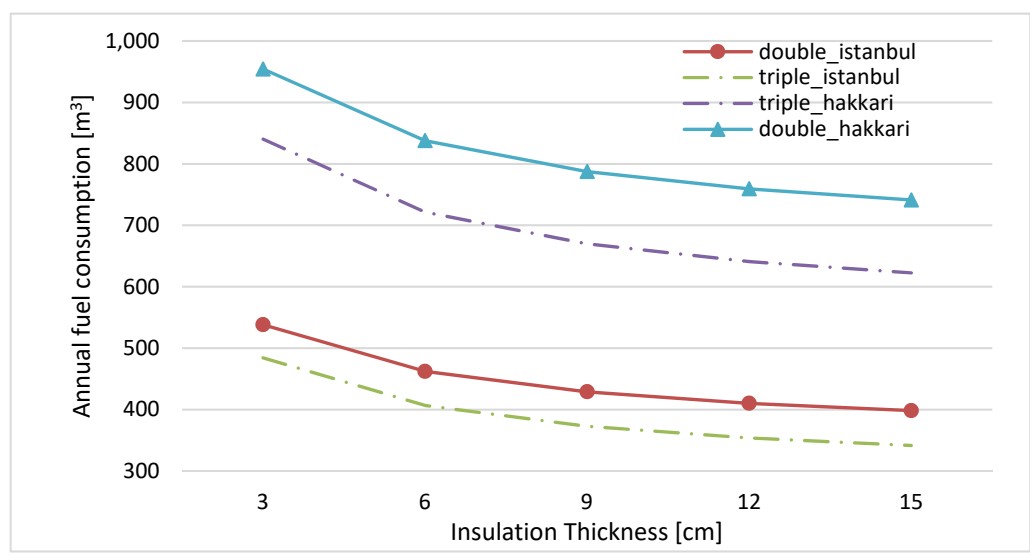

**Figure 13.** Effect of insulation thickness and window type on fuel consumption.

In Figures 14 and 15, the energy savings ratio is presented for varying insulation thicknesses and glazing types. Results show that, especially for single glazed façades, increasing the insulation thickness increases the energy savings ratio at a higher rate in Hakkari compared to Istanbul. Increasing the thermal insulation thickness of the external walls with a triple glazed façade changes the energy savings ratio between 0.25 and 0.48 in both locations. For a double glazed façade, the energy savings ratio changes between 0.20 and 0.41.

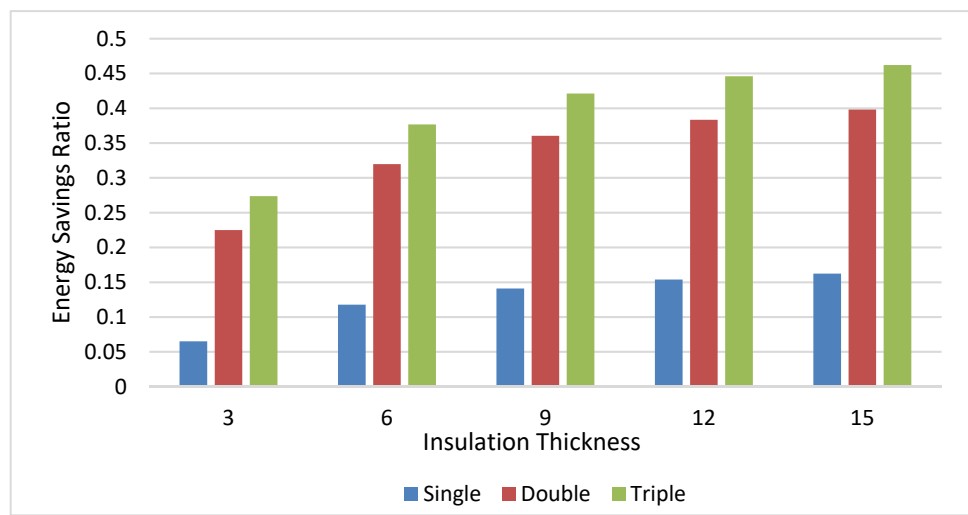

**Figure 14.** Energy savings ratio versus varying insulation thicknesses and glazing types for the designed zone in Hakkari (40% WWR).

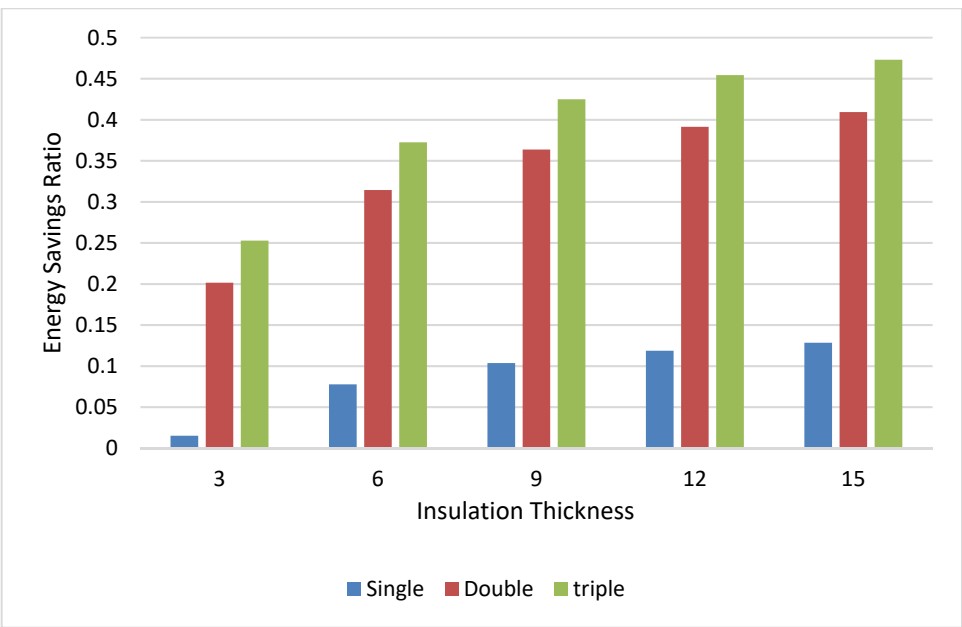

**Figure 15.** Energy savings ratio versus varying insulation thicknesses and glazing types for the designed zone in Istanbul (40% WWR).

### 4.4. Infiltration Rate

Infiltration is the uncontrolled movement of air through the building envelope. Heating load is strongly influenced by the air infiltration rate [17,18]. Infiltration has a negative impact on the heating load as it will leak the heat contained in the building to the outside environment [17]. For the base scenario, a relatively low infiltration rate was applied (0.2 ACH). In this part of the study, the effect of the infiltration rate on the net present cost was investigated. The values of the infiltration rate were incremented between 0.2 and 1 ACH. The results of the simulation process are displayed in Figures 16 and 17 for Istanbul and Hakkari, respectively. It can be seen that varying the infiltration rate from 0.2 to 1 ACH increases the heating energy requirement intensively; therefore, the net present cost rises. Between 0.2 and 1 ACH, the total net present cost increases from 36% to 48% in Istanbul (Figure 16) and 40 to 55% in Hakkari (Figure 17). In both locations, the designed zone favors an air-tight construction; however, due to colder climatic conditions, this is

more apparent in Hakkari. Appropriate solutions to reduce heat loss through infiltration should be carefully considered to keep infiltration at a lower rate.

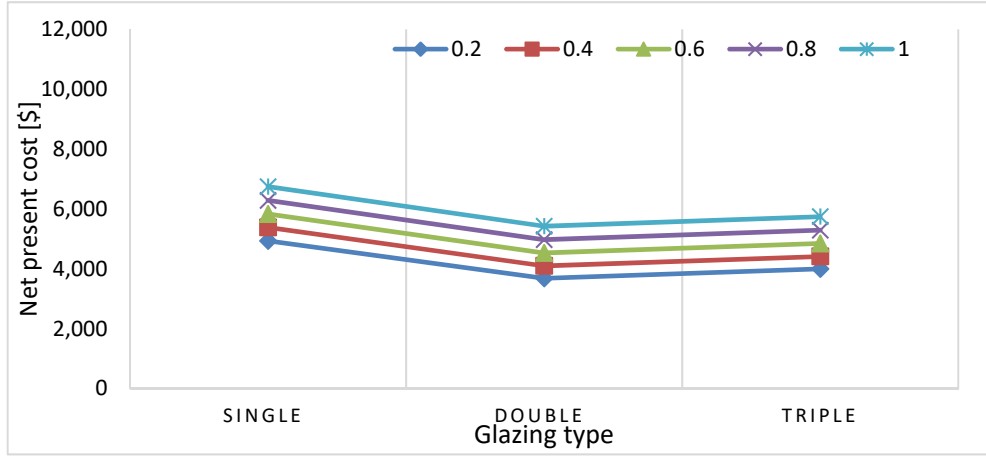

**Figure 16.** The influence of the infiltration rate on the net present cost for Istanbul.

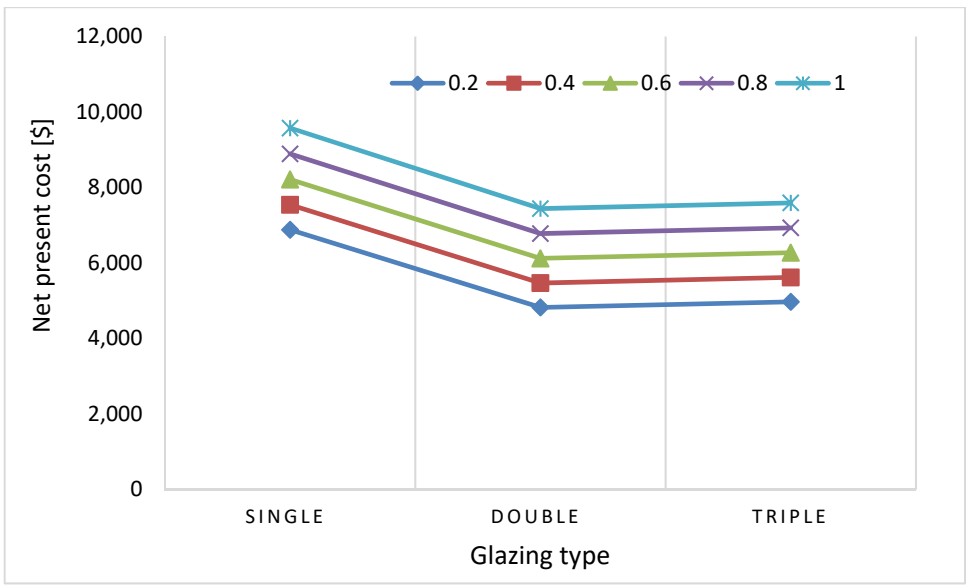

**Figure 17.** The influence of the infiltration rate on the net present cost for Hakkari.

### 4.5. Room Set-Point Temperature

The heating equipment works to bring the zone temperature to the identified set-point temperature. The room set-point temperature impacts the heating load. To see the effect of each 2 °C adjustment, the room set-point temperature was varied from 18 to 26 °C. As depicted in Figures 18 and 19, decreasing the room set-point temperature decreases the annual heating energy consumption; therefore, it reduces the net present cost. This tendency is more visible for the zone with single glazed windows. For a 2 °C change in the temperature, the total net present cost increases from 7% to 15%. Therefore, increasing the occupants' awareness regarding temperature control and obtaining energy control mechanisms based on building occupancy is very crucial for energy economics.

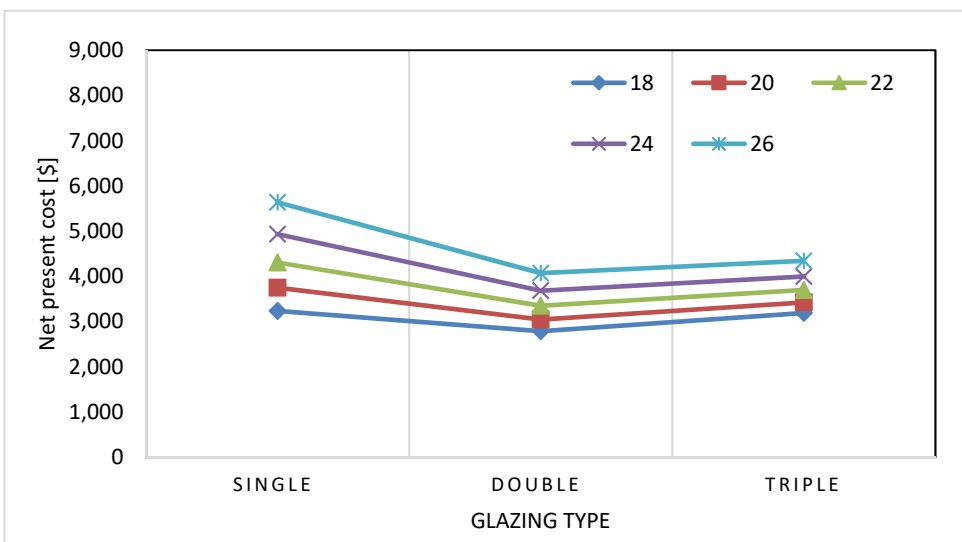

**Figure 18.** The influence of the heating set-point temperature on the net present cost for Istanbul.

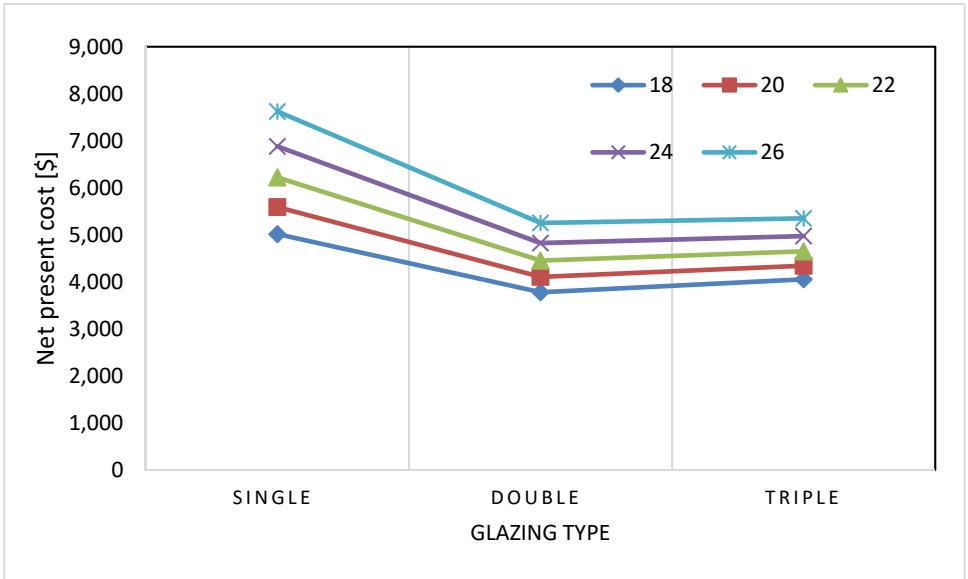

**Figure 19.** The influence of the heating set-point temperature on the net present cost for Hakkari.

*4.6. Fuel Type*

Fuel type has a major impact on the annual heating cost and the net present cost. In Figures 20 and 21, the net present cost of the building envelope is presented for various energy types. Natural gas, electricity, coal, LPG, and fuel-oil are the most widely used energy sources for heating; therefore, they were selected for the parametrical study. The lowest net present cost is for natural gas, followed by coal and fuel oil, respectively. Electricity has the highest net present cost. Since electricity generation in Turkey mostly depends on natural gas and coal, the unit electricity cost is high; therefore, electricity for heating is more expensive than other energy sources. The most suitable energy source for heating was found to be natural gas compared to the other selected energy sources. Preferring natural gas instead of electricity decreases the net present cost by 70% to 85% in both locations, depending on the glazing type.

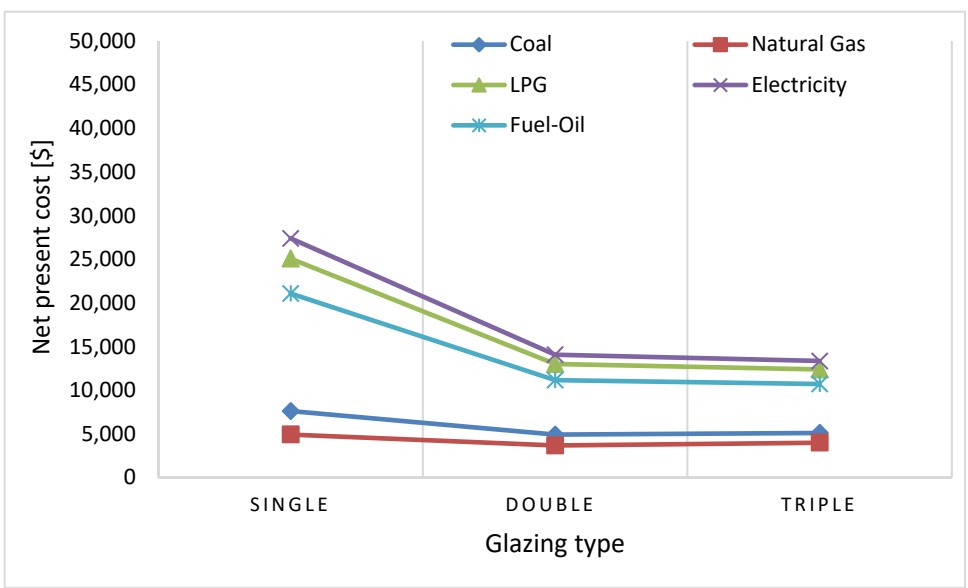

**Figure 20.** The influence of the fuel type on the net present cost for Istanbul.

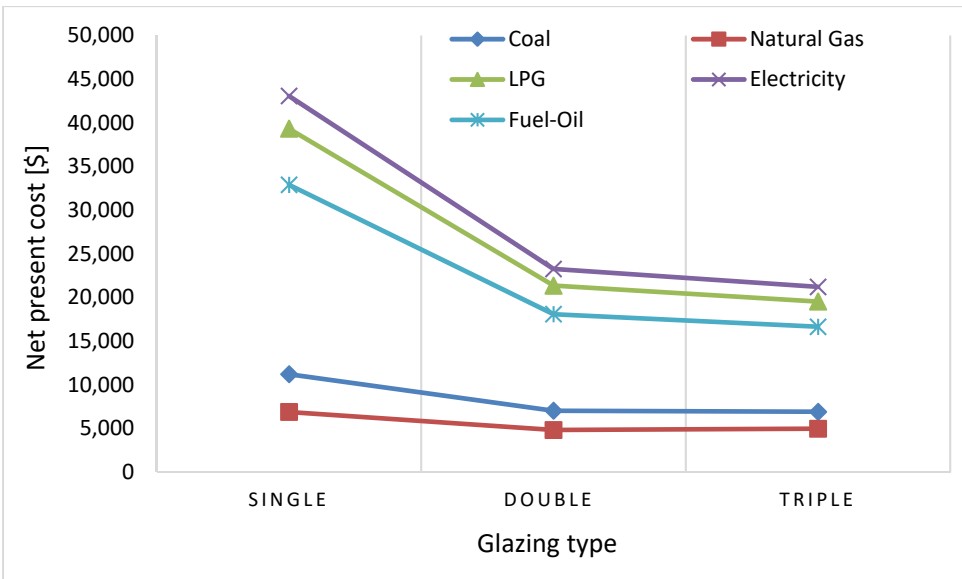

**Figure 21.** The influence of the fuel type on the net present cost for Hakkari.

## 5. Conclusions

The energy efficiency of the buildings is a crucial matter, and mostly depends on early design decisions. Therefore, many have researchers investigated optimum insulation thickness to maximize economic profit and minimize energy consumption. Previous studies have investigated the optimum insulation thickness of the buildings without considering the façade as external walls and windows together. Moreover, parameters such as actual meteorological data (solar radiation, environmental temperature, etc.), infiltration rate, and occupant density influence the heating load of a building; therefore, they should not be neglected. This study evaluated the thermo-economic performance of a zone to determine the optimum façade parameters, such as insulation thickness and window type, using dynamic modelling. The optimum insulation thickness of a zone was chosen over a 20-year lifetime for two different locations. The main results of the study are as follows.

Despite having the lowest initial investment cost, the façade with single glazed windows had the highest net present worth in both locations. The main reason behind that trend is the high cost of fuel due to the more significant heating load for single glazed windows.

It was found that the effect of the façade orientation on the net present cost of the system is significant. Results show that the net present cost of the system can be decreased between 11% and 17% if the orientation of the façade is changed from north-east to south-west.

The surface area of the windows has a significant influence on the net present cost, fuel cost, and the investment cost of the façade. For the façade with single glazed windows, the heating cost increases for greater window areas, and this trend is more dramatic in Hakkari due to colder climatic conditions. For the double glazed and triple glazed windows, the heating energy cost decreases with increasing window areas. However, the net present cost of the façade increases with the growing window area for all cases. For the same window size and window type, in south-oriented façades, the net present cost is always lower than the north-oriented façades because of the passive heating and higher solar radiation levels.

Results show that increasing the insulation thickness of the external walls increases the investment cost and decreases the natural gas consumption. The façade with a single glazed window and 3 cm insulation has the minimum investment cost. The optimum insulation thickness of the building façade was determined based on the minimum net present cost. In Hakkari and Istanbul, the net present cost of the building façade is minimum for 9 cm and 6 cm insulation thicknesses, respectively. The minimum net present cost was found for the façade with double glazed windows in both locations. The optimum insulation thickness of the selected areas can decrease annual fuel consumption by between 14% and 18%.

The infiltration rate is a highly influential parameter for the net present cost of the buildings, especially for the colder climatic regions. To keep infiltration heat losses at the minimum, appropriate solutions should be carefully considered.

Room set-point temperature is another essential parameter for energy-efficient building design. Results of the sensitivity analysis revealed that only a 2 °C change in the temperature increases the net present cost between 7% and 15%. This tendency is more apparent for the single glazed façades due to the greater risk of heat loss.

Finally, sensitivity analysis was also conducted to determine the influence of the fuel type on the net present cost of buildings. The five most common energy resources for heating were evaluated and compared in terms of heating cost and the net present cost. Natural gas was found to be the most suitable energy source for heating, followed by coal and fuel oil. Since national electricity is mainly generated from natural gas and coal, the unit price of the electricity cannot compete with that of coal and natural gas. However, with the broader application of renewable-based power generation systems, it is expected to decrease in the future.

The presented results emphasize the importance of the design parameters, such as orientation, glazing type, insulation thickness, and window area, on the energy efficiency and economy; therefore, designers and engineers should consider all these aspects.

**Funding:** This research received no external funding.

**Institutional Review Board Statement:** Not applicable.

**Informed Consent Statement:** Not applicable.

**Data Availability Statement:** Not applicable.

**Conflicts of Interest:** The author declares no conflict of interest.

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
