# Peer review of "Determination of Optimum Building Envelope Parameters of a Room concerning Window-to-Wall Ratio, Orientation, Insulation Thickness and Window Type"

_buildings, doi:10.3390/buildings12030383_

Round 1

Reviewer 1 Report

  • explain in parentheses what some abbreviations mean
    - page2: COP (
    - table 4: HDD, CDD
  • the description in figure 4 in one row

    In this study, research has been carried out to find optimum building envelope design parameters (insulation thickness, orientation, glazing type, and the window-to-wall ratio of a room) using actual climatic data of two cities (Hakkari and Istanbul) with different characteristics. It was found that appropriate selection of windows,orientation and insulation thickness would lead to a significant reduction of the annual energy consumption.
This study was conducted over a period of 20 years of the building's life.
 A comprehensive parametric evaluation was made by varying the insulation thickness of the external wall,glazing type, glazing area and wall orientation. Parametric evaluation is carried out for a reference room with 100 m2 floor area and with 2 exterior walls. The dynamic simulation results gave important insights for both energy savings and minimizing the net present cost.

    In my opinion, the manuscript’s strengths is given by the use in the study, of window to wall ratio (WWR),because the previous studies investigated optimum insulation thickness of the buildings without considering the façade as external walls and windows together. Results of this study revealed information to design building façade with minimum energy requirement. The methodology used in the study can be applied by other researchers, engineers and architects around the world to design both energy-efficient and cost-effective buildings.

    I don't think they are necessary scientific recommendations for the improvement of the manuscript.

Author Response

First of all I would like to thank you for your constructive comments. I believe that, in the light of your recommendations, the quality of my research is enhanced.

  • Point 1: explain in parentheses what some abbreviations mean
    - page2: COP (
    - table 4: HDD, CDD

Response 1: Thank you very much for your concern. All abbreviations were explained in parentheses in the revised version.

  • Point 2: the description in figure 4 in one row

Response 2: It was decribed in one row in the latest version.

 General comments: In this study, research has been carried out to find optimum building envelope design parameters (insulation thickness, orientation, glazing type, and the window-to-wall ratio of a room) using actual climatic data of two cities (Hakkari and Istanbul) with different characteristics. It was found that appropriate selection of windows, orientation and insulation thickness would lead to a significant reduction of the annual energy consumption.
This study was conducted over a period of 20 years of the building's life.
 A comprehensive parametric evaluation was made by varying the insulation thickness of the external wall,glazing type, glazing area and wall orientation. Parametric evaluation is carried out for a reference room with 100 m2 floor area and with 2 exterior walls. The dynamic simulation results gave important insights for both energy savings and minimizing the net present cost.

    In my opinion, the manuscript’s strengths is given by the use in the study, of window to wall ratio (WWR),because the previous studies investigated optimum insulation thickness of the buildings without considering the façade as external walls and windows together. Results of this study revealed information to design building façade with minimum energy requirement. The methodology used in the study can be applied by other researchers, engineers and architects around the world to design both energy-efficient and cost-effective buildings.

    I don't think they are necessary scientific recommendations for the improvement of the manuscript.

Response 3: I appreciated your kind comments. Thank you very much .

Reviewer 2 Report

This paper determined the optimum design of building envelope including WWR, orientation, insulation thickness and window type for the two representative cities of Turkey by using the normal thermo-economic analysis method.  

However, the analytical methods are not innovative enough. It uses an overly simple building simulation (not reality), and even the process of calculating the heat loads or analyzing the dynamic and transient heat only introduced the use of TRNSYS software and is not detailed enough in this paper. Also, is only natural gas used in these two cities of Turkey? Coal or other energy sources are not introduced and analyzed at all, which is not in line with reality.

In conclusion, I think the research in this article is a bit old and lacks innovation. 

Author Response

(The authors gave the same response as above.)

Reviewer 3 Report

The author conducted this study to determine the most optimal building envelope design characteristics, such as insulation thickness, the orientation of the structure, glazing type, and the room's window-to-wall ratio. It was discovered that the proper selection of windows, orientation, and insulation thickness might result in a considerable decrease in the yearly energy consumption of a home or building.

The following changes are suggested:

  • The heating setpoint is assumed as 24 degC. Is there any basis for this assumption? The same is true for the infiltration rate.
  • In order to support the financial analysis parametric values in Table 3, references must be provided.
  • The monthly temperatures are stated as ‘mean’ in Line 240 (pg 7) and as ‘average’ in Figs 2 and 3 captions. Average is finding the central value in math, whereas mean is finding the central value in statistics. It is unclear whether the author considered mean value or average value in the graphical representation.
  • In all cases, the $ symbol should occur before the actual amount.
  • The rationale for certain graphs being in black and white and others being in colour is unknown. It should be applied consistently across the manuscript.
  • The concluding remarks can simply be stated in paragraphs. Dot points are not required.

Author Response

First of all I would like to thank you for your constructive comments. I believe that, in the light of your recommendations, the quality of my research is enhanced.

General comments: The author conducted this study to determine the most optimal building envelope design characteristics, such as insulation thickness, the orientation of the structure, glazing type, and the room's window-to-wall ratio. It was discovered that the proper selection of windows, orientation, and insulation thickness might result in a considerable decrease in the yearly energy consumption of a home or building.

The following changes are suggested:

  • Point 1: The heating setpoint is assumed as 24 degC. Is there any basis for this assumption? The same is true for the infiltration rate.

Response 1: Thank you very much. You are absolutely showed me a very important point. In the revised form, in addition to base scenario I included additional sections (Section 4.4 for infiltration rate case, Section 4.5 for Room set-point temperature case, Section 4.6 Different fuel types). Therefore, I also studied the influence of the infiltration and the room set-point temperature on the net present cost, in the revised form. Results were highlighted the importance of these two variables. Essential concluding remarks were given as a result of the sensitivity analysis.

  • Point 2: In order to support the financial analysis parametric values in Table 3, references must be provided.

Response 2: Thank you very much. References were included in the revised form.

  • Point 3: The monthly temperatures are stated as ‘mean’ in Line 240 (pg 7) and as ‘average’ in Figs 2 and 3 captions. Average is finding the central value in math, whereas mean is finding the central value in statistics. It is unclear whether the author considered mean value or average value in the graphical representation.

Response 3: Thank you very much. It was corrected.

  • Point 4: In all cases, the $ symbol should occur before the actual amount.
  • Response 4: Thank you very much. It was corrected.
  • Point 5: The rationale for certain graphs being in black and white and others being in colour is unknown. It should be applied consistently across the manuscript.
  • Response 5: Thank you very much. It was corrected.
  • Point 6: The concluding remarks can simply be stated in paragraphs. Dot points are not required.
  • Response 6: Thank you very much. It was corrected.

Round 2

Reviewer 2 Report

Thank you for your effort in revising your paper.

I think the revision has been improved according to my comments. This mainly reflects in that the different energy sources such as coal, LPG, electricity, etc have been added for analysis. Also, the infiltration rate is added for much understood by readers. 

However, the English of paper should be double checked and be corrected carefully.